# Preparation and Investigation of the SPF and Antioxidant Properties of O/W and W/O Emulsions Containing Vitamins A, C and E for Cosmetic Applications

**Nikolaos D. Bikiaris** [1], **Ioanna Koumentakou** [1], **Katerina Hatzistamatiou** [1], **Smaro Lykidou** [1], **Panagiotis Barmpalexis** [2] **and Nikolaos Nikolaidis** [1,*]

1   Laboratory of Chemistry and Technology of Polymers and Dyes, Department of Chemistry, Aristotle University of Thessaloniki, 54124 Thessaloniki, Greece; katerinahatzistamatiou@gmail.com (K.H.)
2   Department of Pharmaceutical Technology, School of Pharmacy, Aristotle University of Thessaloniki, 54124 Thessaloniki, Greece; pbarmp@pharm.auth.gr
*   Correspondence: nfnikola@chem.auth.gr

**Abstract:** In the current work, Oil in Water (O/W) and Water in Oil (W/O) emulsions containing Vitamins A, C and E in 0.5, 1 and 2% wt concentrations were prepared. The pH and viscosity stability over storage, as well as the sunscreen and antioxidant properties of the obtained emulsions, were investigated. The results obtained showed that vitamins slightly increased the pH of the blank emulsions; however, their pH values were within the acceptable values (pH = 4–6). Nevertheless, all emulsions presented excellent pH stability during storage for up to 90 days. Similar results were observed by rheological measurements as the prepared emulsions did not exhibit viscosity instabilities deriving during storage. Moreover, emulsions containing Vitamin A exhibited higher UV protection than the other emulsions, as the W/O emulsion containing 2% wt Vitamin A presented the highest SPF value at 22.6.

**Keywords:** emulsions; oil in water (O/W); water in oil (W/O); vitamin A; vitamin E; vitamin C; UV protection; antioxidant properties

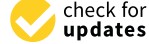



## 1. Introduction

Skin is the primary defensive barrier of the human organism against pathogens and chemical threats. It comprises three main layers: the stratum corneum (SC), the dermis (DE) and the hypodermis [1–3]. Ultraviolet (UV) irradiation, naturally abundant in the environment through sun exposure, subsidizes a variety of skin maladies, including degenerative aging, inflammation and cancer. Moreover, UV exposure is responsible for the generation of reactive oxygen species (ROS) that may have detrimental effects on the skin's health. In addition, a great number of active substances have been applied to the surface of the skin for therapeutic and pharmaceutical purposes through cosmetic products, such as body, face and hand emulsions [4–6].

Emulsions are complex formulations used for optimizing various end-use properties of dermatological substances, such as the stability and delivery of active compounds [7]. The basic formulation of a standard emulsion contains at least water as the continuous phase, oil as the dispersed phase, and emulsifiers ensuring the stability of the dispersion of oil droplets [8]. In the past, oil in water (O/W) and water in oil (W/O) emulsions have been extensively used and studied for the dermal delivery [9] of numerous active ingredients such as vitamins [10,11], retinoids [12], drugs [13], proteins and peptides [14].

Vitamins, one the most-known antioxidants used in the cosmetics industry, are naturally occurring in human skin, where they prevent lipid peroxidation at the cell membrane level, stabilizing especially those membranes with high content of polyunsaturated fatty acids [15]. Vitamin A (Vit A) is a generic term for a family of organic lipid-soluble retinal

that is essential for the performance of multiple biological functions, such as normal growth and development [16]. The most effective forms of Vit A are retinal and retinol, which contain an aldehyde or an alcohol end group, respectively [17]. Vitamin C is a major antioxidant in the aqueous cell compartment with the capability to neutralize oxidative stress generated by various factors, with the most common source being UV exposure [18,19]. In certain skin disorders, the use of ascorbic acid (Vit C) can create an acidic environment that contributes to wound healing by regulating the wound infection and barrier function, increasing the antimicrobial activity and enhancing epithelization and angiogenesis [20]. Vitamin E (Vit E) can be found in high concentrations in the deepest layers of the stratum corneum (SC), forming the primary defense of the skin to the oxidative stress induced by UV exposure. Topical delivery of Vitamin E protects the skin against UV-caused cutaneous harm, and carcinogenic and mutagenic activity of chemical agents [21,22].

Yoshida, et al. [23], proposed the preparation of O/W/O emulsions for the stabilization of Vitamin A. Maia, et al. [24] investigated the influence of sodium metabisulfite and glutathione antioxidants on the stability of Vitamin C in O/W emulsions. It was found that the storage condition of $40.0 \pm 0.5$ °C accelerated the degradation rate of Vitamin C, despite the presence of the antioxidants. Fraj, et al. [25] studied the in vitro release of Vitamins C and E from microcapsules stabilized by gelatin sodium caseinate system in double W/O/W emulsions. Moyano and Segall [26] evaluated the stability of Vitamin A palmitate in the presence of Vitamin E and other antioxidants in lipophilic/hydrophilic medium (O/W emulsions) at pH 3.0, 5.0 and 7.0.

In the current work, a series of O/W and W/O emulsions containing Vitamins A, C and E have been prepared, and their sunscreen activity and their antioxidant properties are studied for the first time.

## 2. Materials and Methods

### 2.1. Materials

Vit A (Retinyl palmitate), Vit C (L-Ascorbic acid) and Vit E (Alpha-Tocopherol) were received by Tzimas Cosmetics (Thessaloniki, Greece). For the preparation of O/W and W/O emulsions, olive oil, shea butter, glycerin, cetearyl alcohol, cetyl alcohol, sodium citrate, beeswax, xanthan gum, polysorbate 60 (Tween 60), stearic acid, ethylhexylglycerin and phenoxyethanol were purchased by Novita Group (Thessaloniki, Greece). All other materials and reagents used in this study were of analytical grade of purity.

### 2.2. Preparation of the O/W and W/O Emulsions

The preparation of the O/W and W/O emulsions was conducted according to a technique described in previous works [27,28]. Briefly, Table 1 summarizes the two phases and the utilized substances for the emulsions' fabrication. In summary, the water phase was completely dissolved in a water bath at 70 °C. The oil phase was heated separately in a water bath at 70 °C until all the substances were dissolved and were then added dropwise to the water phase under constant mechanical stirring (600 RPM) using a 2020 RZR (Heidolph, Schwabach, Germany) mechanical stirrer until complete homogenization. The prepared emulsions were left under mechanical stirring (400 RPM) for 1 h; vitamins were added at 45 °C in various weight ratios (0.5, 1 and 2% $w/w$) as illustrated in Table 2, and mechanical stirring continued for 1 more hour. Afterwards, ethylhexylglycerin and phenoxyethanol were added as preservatives at room temperature, and mechanical stirring was continued for 30 min. Regarding the O/W formulations, the water phase and the oil phase of the final emulsions were 25% $w/w$ and 75% $w/w$, respectively. On the other hand, W/O emulsions contained 75% $w/w$ of the oil phase and 25% $w/w$ of the water phase. Lastly, two blank emulsions (O/W and W/O) without the addition of any vitamins were also prepared as control samples. The final volume/weight ratio of the produced emulsions was 150 mL/200 g. The fabricated formulations were stored in sterilized glass sample jars (250 g) at room temperature.

**Table 1.** Water and oil phases for the preparation of the O/W and W/O emulsions.

| Water Phase | | |
|---|---|---|
| **Ingredients** | **O/W Emulsions (75%)** | **W/O Emulssions (25%)** |
| Water | 70 gr | 23.5 gr |
| Glycerin | 3.5 gr | 1.15 gr |
| Citric acid | 0.5 gr | 0.175 gr |
| Xanthan gum | 1 gr | 0.175 gr |
| **Oil phase** | | |
| **Ingredients** | **O/W emulsions (25%)** | **W/O emulsions (75%)** |
| Olive oil | 11 gr | 43.75 gr |
| Cetyl alcohol | 2 gr | 3.25 gr |
| Cetearyl alcohol | 2 gr | 3.25 gr |
| Stearic acid | 2 gr | 6.5 gr |
| Shea butter | 2 gr | 6.5 gr |
| Beeswax | 2 gr | 6.5 gr |
| Tween 60 | 2 gr | 3.25 gr |

**Table 2.** Abbreviations of the prepared emulsions and their contents in Vitamins A, C and E.

| | 0.5% Vit A O/W | 1% Vit A O/W | 2% Vit A O/W | 0.5% Vit A W/O | 1% Vit A W/O | 2% Vit A W/O | 0.5% Vit C O/W | 1% Vit C O/W | 2% Vit C O/W | 0.5% Vit C W/O | 1% Vit C W/O | 2% Vit C W/O | 0.5% Vit E O/W | 1% Vit E O/W | 2% Vit E O/W | 0.5% Vit E W/O | 1% Vit E W/O | 2% Vit E W/O |
|---|---|---|---|---|---|---|---|---|---|---|---|---|---|---|---|---|---|---|
| Vitamin A | 0.5 gr | | | 0.5 gr | | | 0.5 gr | | | 0.5 gr | | | 0.5 gr | | | 0.5 gr | | |
| Vitamin C | | 1 gr | | | 1 gr | | | 1 gr | | | 1 gr | | | 1 gr | | | 1 gr | |
| Vitamin E | | | 2 gr | | | 2 gr | | | 2 gr | | | 2 gr | | | 2 gr | | | 2 gr |

### 2.3. pH Measurement

The stability of the prepared emulsions was studied via pH and viscosity measurements after 1, 7, 14, 30, 60 and 90 days of storage in RT after the preparation of the emulsions [29]. The pH measurements of the emulsions were read precisely from the instrument by dipping the pH sensor (Microprocessor, WTW, pH 535, Gemini BV, Apeldoorn, The Netherlands), and the mean value of three consecutive measurements was calculated. Variations in the results of pH measurements could not exceed 10% of the initial value for the emulsion to be considered stable.

### 2.4. Viscosity Determination

Viscosity measurements were performed using the R3 spindle of a Visco Star Plus viscometer (Controla S.A, Thessaloniki, Greece) under 50 and 100 rpm after 1, 7, 14, 30, 60 and 90 days of storage in RT after the fabrication of the emulsions. Apparent viscosity determinations were assessed by dipping the R3 spindle of the viscometer for 1 min in each of the studied emulsions [30], and the obtained values were expressed as Centipoise (cP). Variation in the results of viscosity measurements could not exceed 10% of the initial value for the emulsion to be considered stable.

### 2.5. Sun Protection Factor (SPF)

The diluted solution transmittance method was employed for the investigation of the SPF values of the emulsions. Briefly, 1% $w/v$ from each sample was added in ethanol, completely homogenized using sonication and filtered through Whatman filters. 20% $v/v$ of each solution was diluted with ethanol. The absorption values of the emulsions were achieved in the range of 290–320 nm (every 5 nm) using a UV-Vis spectrophotometer

(Shimadzu, Tokyo, Japan) [28]. Mansur equation was utilized to determine the SPF values of the emulsions:

$$\text{SPF in vitro} = CF \ast \sum_{290}^{320} EE(\lambda) \ast I(\lambda) \ast abs(\lambda) \tag{1}$$

where CF (correction factor) is 10), EE($\lambda$) is the erythemogenic effect of radiation at wavelength $\lambda$, I($\lambda$) denotes the intensity of solar light at wavelength $\lambda$ and abs($\lambda$) denotes the absorbance of the sample at wavelength $\lambda$. "EE" and $\times$ "I" are constants that were first determined in the work of Sayre, et al. [31].

### 2.6. Antioxidant Study

The antioxidant activity (AA) of the prepared emulsions was determined via the 2,2-Diphenyil-1-picrylhydrazyl (DPPH) method, according to the method developed by Blois in 1958 [32]. Specifically, 1 mL of each sample was dispersed in ethanol (1% *v/v*) and was added to 3 mL of a $5 \times 10^{-3}$ mg/mL ethanol DPPH solution. The reference sample was composed of ethanol and the DPPH/EtOH solution in a 1/3 ratio. The samples were sonicated for 10 min, and their absorbance was recorded after 30 min at 517 nm using a UV-Vis spectrometer (UV Probe 1650, Shimadzu, Tokyo, Japan). The free radical scavenging activity was described according to the following equation [33]:

$$\text{Free radical scavenging activity (\%)} = \frac{\text{Absorbance of control} - \text{Absorbance of extracts}}{\text{Absorbance of control}} \ast 100 \tag{2}$$

All experiments for each formulation were performed in triplicate for the validation of the obtained results.

## 3. Results

### 3.1. pH Stability

Figure 1 presents the obtained pH values for each investigated emulsion during storage at room temperature after 1, 7, 14, 30, 60 and 90 days after their preparation. The pH stability, in general, was excellent, and this can be attributed to the citric acid used as a pH regulator. Obviously, the composition of the emulsion affects the pH. It is well known that olive oil possesses more acidic pH than deionized water; hence W/O type of emulsion is expected to exhibit higher values of pH. All other emulsifiers used in the current work have a neutral pH of around 7 and thus do not have a significant effect on the stability. Generally, the pH values of the studied samples varied in the range of 4–6. However, emulsions with 2% *w/w* Vitamin C exhibited pH values below 4. Consecutively, the specific emulsions are not appropriate for dermal application. For this reason, no further characterization tests (viscosity, SPF, antioxidant) were performed on these samples.

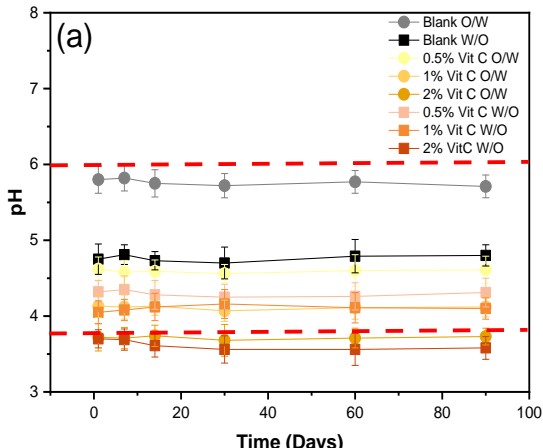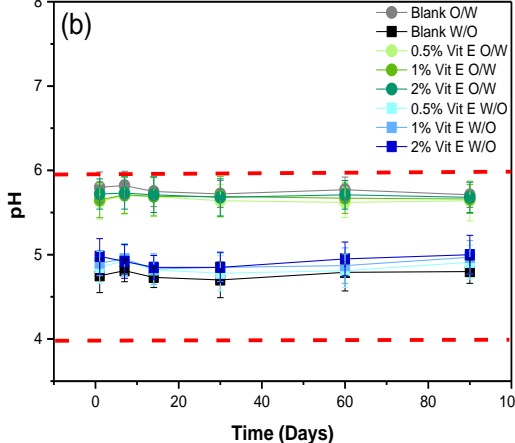

**Figure 1.** *Cont.*

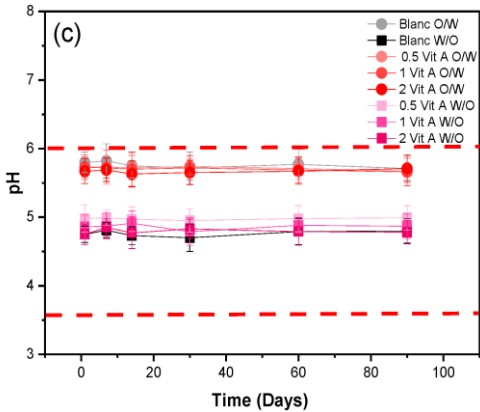

**Figure 1.** Change in pH values over time for O/W and W/O Blank emulsions, O/W and W/O emulsions with (**a**) Vitamin C; (**b**) Vitamin E; and (**c**) Vitamin A.

*3.2. Rheological Study*

Viscosity measurements were determined for all the prepared emulsions, and their stability profiles at 50 and 100 RPM during storage for up to 90 days are presented in Figures 2–4. It is clearly evident that no viscosity instabilities were observed during storage. This is due to the nature of the emulsifiers and the viscosity regulators used in the water and in the oil phase. The xanthate gum used in the water phase is an excellent polysaccharide-based viscosity stabilizer and thickening agent, producing very stable emulsions. Additionally, the Tween 60, possessing 60 moles of the long hydrophilic ethylene oxide chain, confers excellent emulsifying properties, thus stabilizing the emulsion. The increased amount of oily and fatty substances, such as oil and beeswax, resulted in increased viscosity values of the W/O blank. This is a result of the stickiness of fat particles, and the viscosity increase is attributed to the solidification of fat. Moreover, both at 50 and 100 RPM, similar values with the blank sample, varying in the range of 30–80 cP, were obtained for all O/W formulations containing the studied vitamins, with the ones at 50 RPM being higher. Regarding the W/O emulsions, significant changes were noticed. Specifically, at 50 RPM, Vit A samples showed decreased viscosity compared to blank W/O, with the one containing 0.5% Vit A having the lowest value around 200 cP. On the contrary, at 100 RPM, emulsions with 1% Vit A showed higher values compared to the blank sample. Vit C formulations, both at 50 and 100 RPM, exhibited similar and higher values than the blank W/O cream. Lastly, all samples containing Vit E exhibited higher values than the control sample. At 50 RPM, the viscosity profiles of all creams were around 350 cP. However, at 100 RPM, 0.5% Vit A sample was near 175 cP. 1 and 2% Vit A samples exhibited increased values near 300 cP.

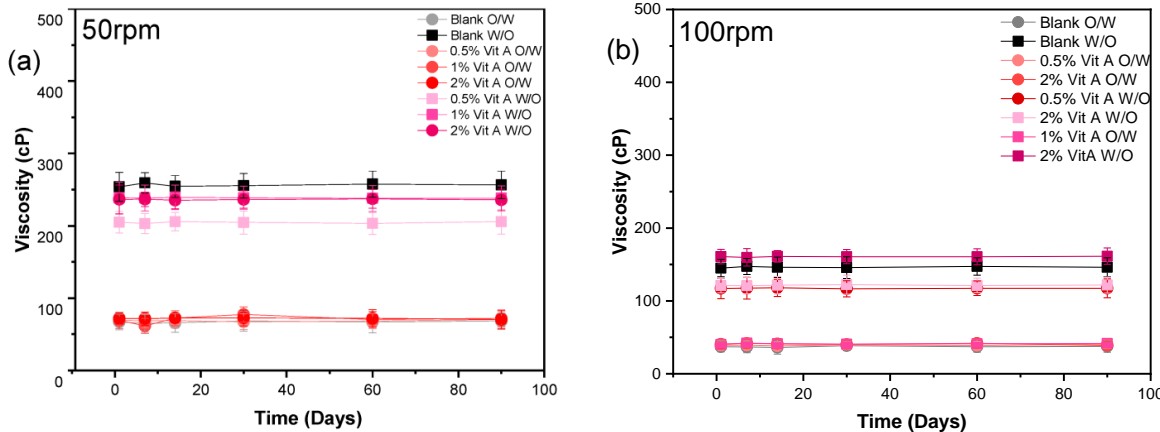

**Figure 2.** (**a**) Viscosity tests for Vitamin A emulsions at (**a**) 50 rpm and (**b**) 100 rpm.

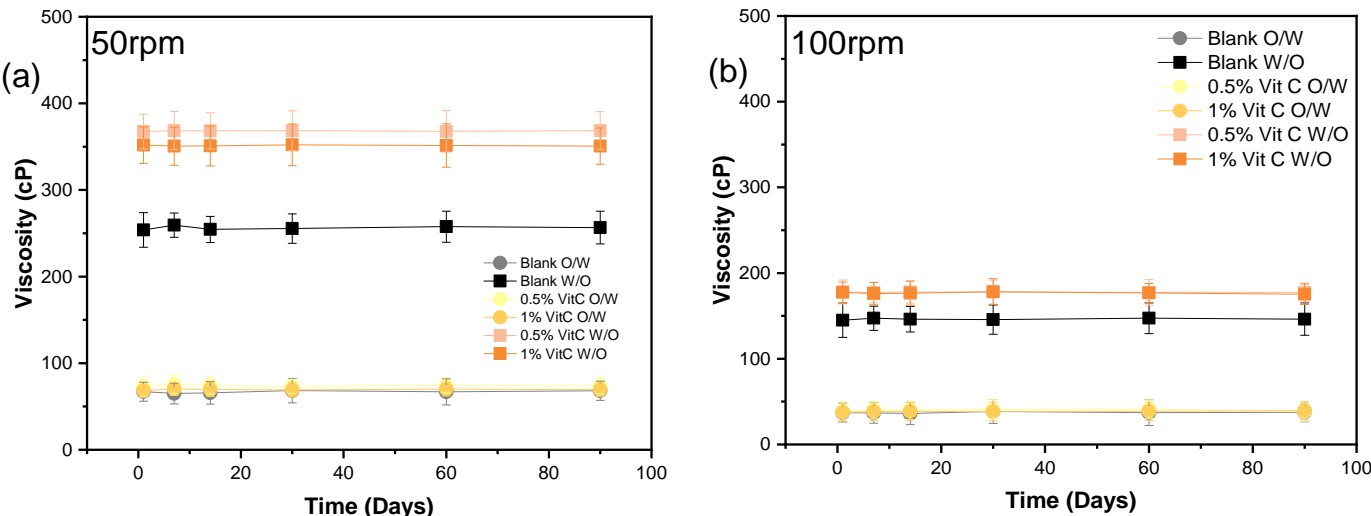

**Figure 3.** Viscosity tests for Vitamin C emulsions at (**a**) 50 rpm and (**b**) 100 rpm.

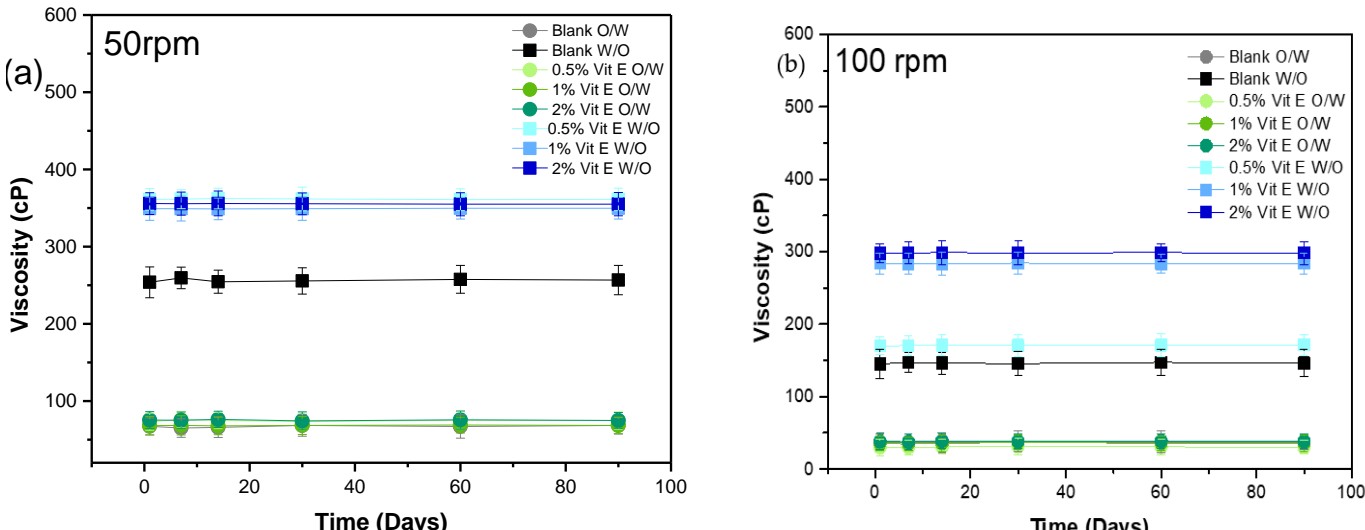

**Figure 4.** Viscosity tests for Vitamin E emulsions at (**a**) 50 rpm and (**b**) 100 rpm.

### 3.3. Assessment of Sunscreen Activity

Figure 5 presents the in vitro SPF tests conducted on the prepared emulsions. Blank O/W and W/O samples exhibited 3.72 and 3.96 SPF, respectively, with the W/O having slightly higher SPF, probably due to the increased amount of olive oil that is well-known for its UV-blocking properties [34]. O/W and W/O emulsions containing Vit A exhibited an increasing trend with increasing vitamin addition. The highest SPF value both amongst the Vit A and the overall formulations was shown by 2% Vit A W/O cream at 22.6. O/W 0.5 and 1% Vit C emulsions exhibited lower SPF values than the blank samples, indicating that the addition of Vit C has a negative effect on the anti-UV behavior. However, the addition of Vit C in W/O emulsions increased the SPF values up to 8.52 in the 1% Vit C W/O sample. A similar value (8.11) was obtained by the 1% Vit E W/O emulsion. The addition of 2% Vit E in both O/W and W/O samples seems to have an inhibitory effect when compared to 1% samples since an SPF decrease is observed.

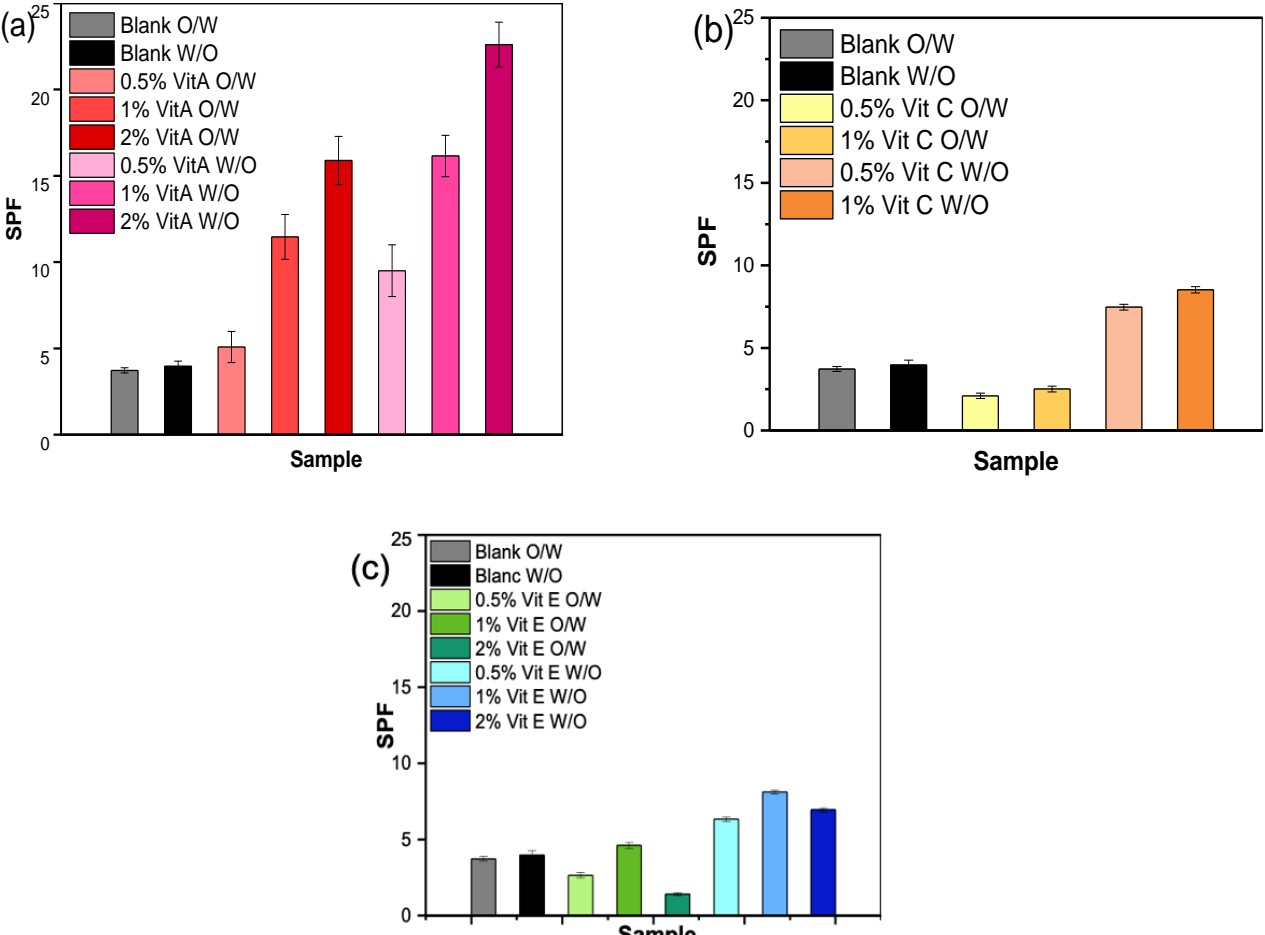

**Figure 5.** In vitro SPF measurements of O/W and W/O emulsions containing (**a**) Vitamin A, (**b**) Vitamin C, and (**c**) Vitamin E.

### 3.4. Antioxidant Properties

Figure 6 illustrates the in vitro antioxidant ability of the prepared emulsions. It was obtained that all W/O emulsions showed enhanced antioxidant capacity compared to O/W emulsions. This is due to the higher amount of olive oil [35] and shea butter [36] that present strong antioxidant properties. All samples containing vitamins exhibited improved antioxidant ability compared to the blank emulsions, which was enhanced with the increase of vitamin content from 0.5 to 2% $w/w$. However, the samples showed significant differences from each other in their antioxidant ability. Emulsion with 1% Vit C W/O exhibited the highest antioxidant capacity at 80.56% AA during the 8 days of this study. The emulsions containing Vitamin E exhibited the next-best results, as 2% Vit E W/O had 77.86% AA. The formulations with Vitamin A showed lower antioxidant capacity since the best result was shown by 2% Vit A W/O cream with 65.23% AA. All emulsions showed a progressive increase in the % antioxidant capacity up to 8 days after their preparation. This was also observed in previous works that studied the antioxidant properties of emulsions containing curcumin derivatives [27]. This may be attributed to the synergistic effect taking place during storage between vitamins and antioxidant ingredients (olive oil and shea butter) used in the oil phase of the prepared emulsions [37].

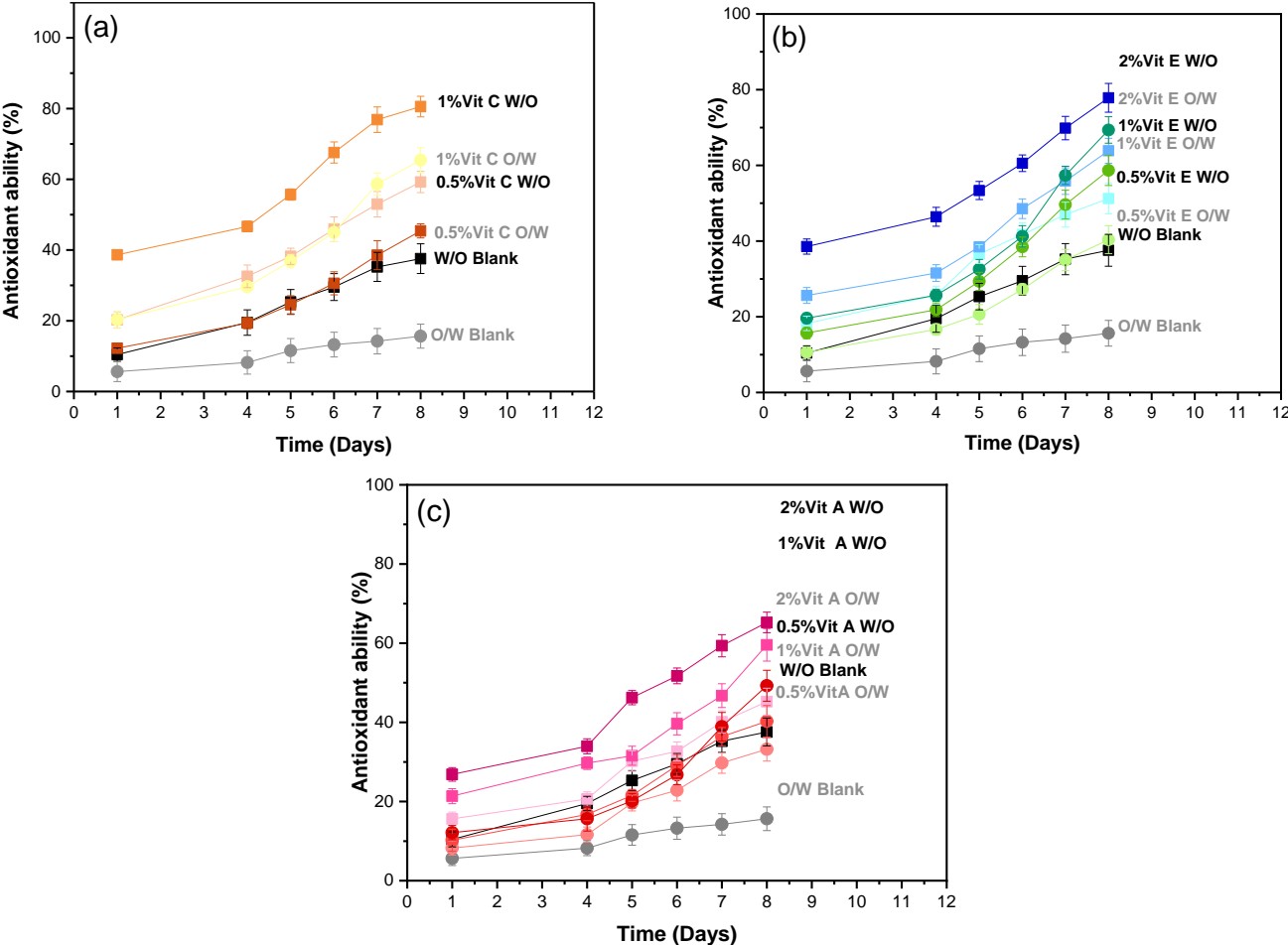

**Figure 6.** Study of the kinetic of the antioxidant ability of emulsions during the first 8 days. (**a**) represents the % antioxidant ability of the emulsions containing Vit C, (**b**) the emulsions containing Vit E and (**c**) the emulsions containing Vit A.

## 4. Discussion

The popularity of a cosmetic product varies not only on the efficiency of the active ingredient(s) but also on consumer approval, which is greatly affected by the sensory properties of the formulation [38]. The physical stability of the creams is an important parameter in the stabilization of the vitamins for topical dermal delivery and for ensuring the effectiveness of their usage. The stability of the formulation, temperature variations during transportation or storage and pH changes may lead to the degradation of the active compounds [21]. Consecutively, these factors in a formulation are critical and should be carefully investigated and monitored to determine the bio-functionality of a cosmetic product and to fulfill consumers' needs.

### 4.1. pH Stability

Since the beginning of its study, human skin has been believed to have an acidic surface [39]. In general, it was proposed by researchers that the skin pH varies between 5.4 and 5.9. Recent works have claimed that the pH of adult skin has an even more acidic nature, with the scientific community accepting that, depending on the part of the body, it ranges from 4.1 to 5.8 (mean 4.9) [40]. In pH values higher than 6, several biological functions of the SC, such as the formation of epidermal lipids and the keratinocyte differentiation process, are suspended, while the development of microbes is favored, resulting in skin disorders and disturbances [41]. Cosmetic formulations, due to their extensive daily application, should contribute to skin health conservation by maintaining the pH of the

skin's surface [42]. Moreover, the pH of a topical product is vital for optimal bioavailability, as is the presence of a penetration enhancer. Taking into consideration the abovementioned, it is of great significance for a topical skin care product to retain pH values between 4 and 6.

In the current work, it was observed (Figure 1) that the pH values of almost all the prepared emulsions remained between 4 and 5.9, even after 90 days of storage at room temperature. These results demonstrate the stability of the creams in terms of pH values even after a long period of storage. Only the emulsions containing 2% *w/v* Vit C, both O/W and W/O, exhibited highly acetic values in the range of 3.1–3.8. As previously mentioned, the pH of a cosmetic product should vary in the range of 4–5.8; otherwise, it could cause skin irritations or suspension of biological functions. This could be attributed to the increased amount of ascorbic acid (Vit C) that was added to each emulsion. Similar results were reported by Cefali, et al. [43], who studied the stability of emulsions containing Vitamin C in acerola extracts. During the stability tests, the pH values of the emulsions dropped to $4.02 \pm 0.3$ due to the fact that Vit C is prone to thermal degradation at elevated temperatures and that light could decrease pH values during storage. Kim, et al. [44] also reported very low values of pH and a constant decrease (from 5 to 2) when studying the stability of L-ascorbic acid in O/W emulsions at 25 °C.

### 4.2. Rheological Study

The viscosity measurement of an emulsion may provide significant information on an emulsion's stability and inner microstructure and, thus, is an important factor to take into consideration. Any variations in viscosity through storage may result in several defects in the aesthetic appearance of the end product [27]. However, from the obtained values of viscosity (Figures 3–5), no apparent viscosity instabilities deriving from the addition of the vitamins are observed. This is a great indication of the excellent viscosity stability of the studied emulsions during storage in RT for up to 90 days. As expected, due to the increased number of oily substances and thick liquid nature, W/O emulsions exhibited very high values of viscosity. Moreover, in the measured samples, the viscosity dramatically decreased with increasing RPM, as shown in Figures 3–5, suggesting that the W/O emulsions exhibit a shear-thinning behavior as pseudoplastic fluids. This thixotropic behavior is highly necessary for topical skin formulations as they are considered to deform during application and become fluid, thus facilitating the spreading of the product on the surface of the skin [45]. Moreover, the recovery of the initial viscosity after application prevents the product from dripping [46].

### 4.3. Assessment of Sunscreen Activity

Exposure to UV irradiation is responsible for numerous destructive effects on human skin health, such as sunburn (erythema), blistering (edema), photoaging and carcinogenesis [47,48]. Thus, when considering the preparation of a cosmetic product, anti-UV properties should be employed. The activity of a sunscreen product is mostly rated and marketed by SPF, which measures the fraction of sunburn. This term may be explained as the fraction of the UV radiation required to induce nominal erythema on the skin after applying sunscreen to the amount of energy necessary to convey the same effect on the skin without any sunscreen. The higher the SPF, the higher the protection created against the sun's rays [49,50]. There are several factors affecting the determination of SPF values, such as the nature of the emulsion, the effects and interactions of vehicle components—such as emollients and emulsifiers inside the product as well as with the skin—the addition of any active ingredients, the pH system and the emulsion rheological properties, among other factors, which may decrease or enhance the UV absorption of a sunscreen formulation [51]. At present, a well-known method in the cosmetic industry is to incorporate antioxidants, such as vitamins [47], as additions to UV filters since almost all post-radiation reactions involve reactive oxygen species (ROS) [52]. These photo-oxidative reactions initiate the development of many disorders affecting the skin [53,54].

The in vitro assessment of SPF of the investigated samples in the current work showed that in all cases, W/O emulsions exhibit higher UV protection (higher SPF). This can be attributed to the presence of an increased amount of olive oil and oily substances. Several scientific works have highlighted the anti-UV efficacy of olive oil in the past [34,55,56]. Kaur, et al. [51] calculated an in vitro 7.549 SPF value using a UV-Vis spectrophotometer. This was the highest value amongst a variety of investigated natural oils such as coconut, almond, castor, peppermint, lavender and lemon grass oil. Regarding the investigated samples, W/O emulsions containing Vit with a 2% ratio exhibited a 22.6 SPF, the highest value amongst all creams. As previously reported by Antille, et al. [57], due to its physical properties, Vitamin A (retinyl esters and retinol) strongly absorbs ultraviolet radiation between 300 and 350 nm, with a maximum at 325 nm. This wavelength range is transmitted from the sun at the earth's level and is accountable for many of the deleterious biological effects of the sun. In a study conducted on mice, it was shown that the topical delivery of Vitamin A prevents UV-induced epidermal hypovitaminosis A, while topical Vitamin E inhibits oxidative stress and systemic immuno- suppression elicited by UV [58]. In the current work, an 8.11 SPF value was obtained by the 1% Vit E W/O emulsion. Additionally, an inhibitory influence was observed with the addition of 2% Vit E in both O/W and W/O samples when compared to 1% samples since SPF noticeably decreased. O/W 0.5 and 1% Vit C emulsions exhibited lower SPF values than the blank samples. This may be attributed to the composition of the emulsion vehicle in combination with the addition of Vit C [59]. Nevertheless, 1% Vit C W/O exhibited SPF values up to 8.52, indicating a potential UV-blocking ability.

### 4.4. Antioxidant Capability of Emulsions

A number of oxygen-derived free radicals (molecules with an unpaired electron, ROS) are induced through many endogenous sources, e.g., enzyme activity or activated neutrophils, and external sources, such as exposure to pollutants, drugs and solar ultraviolet radiation. Superoxide radicals [$O_2 \cdot-$] and hydroxyl radicals [$OH \cdot$] are the most found free radicals in the body. These agents can cause structural alterations to DNA, lipid peroxidation, induce direct damage to lipids and proteins, modify the enzyme structures, secretion of inflammatory cytokines, and finally cause cell death. Free radicals are responsible for at least part of the degenerative changes leading to cutaneous aging and skin cancer [60]. The production of endogenous antioxidant compounds by the human body can protect against ROS. Antioxidants are molecules that stabilize free radicals by the donation of an electron without also becoming free radicals themselves (Figure 7). Their function is to prevent or delay the cellular damage induced by ROS and work by reducing local oxygen concentrations and impairing chain initiation reactions [61]. Nevertheless, the intake of external antioxidant agents that intercept and scavenge radicals to inhibit the initiation and break chain propagation can further enhance protection against free radicals [62,63].

Vitamin C, Vitamin E and Vitamin A are capable of attenuating the damaging effects of $O_2$ in lipid and nonlipid cellular sections, thereby acting as potential antioxidants [64]. Several reports have proposed novel therapeutic approaches with topical or systemic administration of these antioxidant agents [65,66]. According to these results, the emulsions with Vit C exhibited the highest antioxidant ability. Vitamin C can either react directly with chain-carrying peroxyl radicals (ROO', where R = H, substituted alkyl, etc.) or indirectly, by the reduction of the alpha-tocopherol radical to renew Vitamin E [67,68]. Furthermore, it was found that the addition of Vit E (alpha-tocopherol) improved the antioxidant capacity of the prepared emulsions since it is also a potent biological chain-breaking antioxidant and effectively inhibits the peroxidation of lipids [69–71]. However, despite the fact that it has been reported that Vitamin A and its congeners act as chain-breaking antioxidants in biological membranes, in the present work, the addition of Vitamin A illustrated a lower antioxidant rate compared to formulations with vitamins C and E [64].

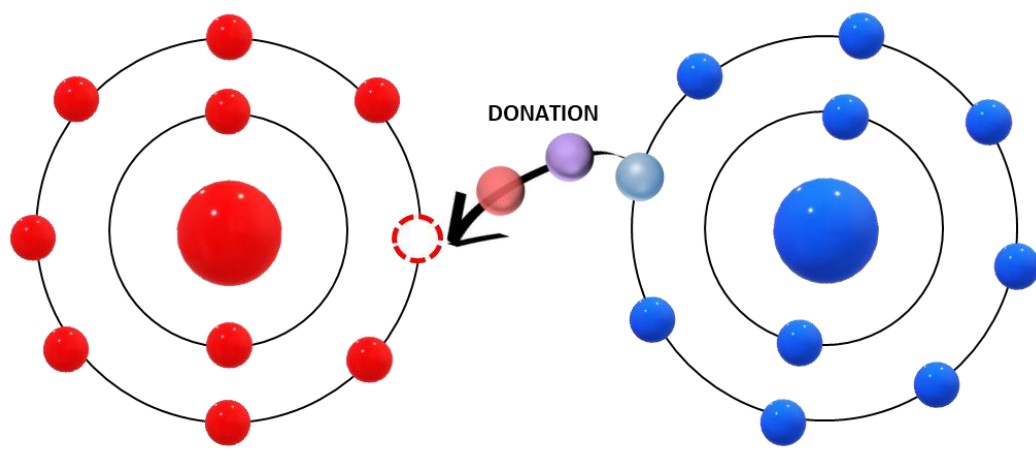

**Figure 7.** The mechanism of antioxidant agents' activity by the donation of an electron to a free radical.

## 5. Conclusions

In this study, O/W and W/O emulsions containing 0.5, 1 and 2% $w/w$ Vitamins A, C and E for topical skin delivery applications are reported. Investigation of viscosity and pH stability at room temperature showed no deviations from the initial values and thus confirmed the stability of the samples over storage. 2% Vit C O/W and W/O samples exhibited pH values below 4, which is the acceptable value for human skin application, and hence their use is prohibited for dermal usage. In vitro SPF determination showed no insignificant values, suggesting that incorporated vitamins exhibit a mild sun protection activity. Great results were obtained by the performed antioxidants test, where a sample with 1% Vit C O/W presented the highest antioxidant ability, reaching up to 80.56% in the first 8 days. This was attributed to the excellent capability of vitamins to scavenge ROS such as $[O_2 \cdot -]$ and $[OH]$. All in all, the results obtained here are promising for the treatment of various skin disorders. Certainly, additional drug release and in vivo studies should be performed in future research since these emulsions are destined for human treatment. Lastly, it would be worth studying complex formulations that contain more than one vitamin for stability over time.

**Author Contributions:** Conceptualization, N.N.; Investigation, K.H. and S.L.; Methodology, S.L.; Project administration, N.N.; Resources, S.L.; Supervision, N.N.; Writing—original draft, N.D.B.; Writing—review and editing, I.K. and P.B. All authors have read and agreed to the published version of the manuscript.

**Funding:** This research received no external funding.

**Institutional Review Board Statement:** Not applicable.

**Informed Consent Statement:** Not applicable.

**Data Availability Statement:** Not applicable.

**Conflicts of Interest:** The authors declare no conflict of interest.

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
