# Peer review of "Preparation and Investigation of the SPF and Antioxidant Properties of O/W and W/O Emulsions Containing Vitamins A, C and E for Cosmetic Applications"

_cosmetics, doi:10.3390/cosmetics10030076_

Round 1
Reviewer 1 Report (Previous Reviewer 2)
The Figures should be uniformized by using the same scale on the y axis. Fig 3, Fig 4, Fig 5 and Fig 6 should be corrected. This important for the comparison of the different Vitamins.
Vitamin E should be mentioned as alpha-tocopherol, not only tocopherol.
The number of references is very high, the not necessary citations should be skipped.
The English language should be checked and polished over the manuscript.
Author Response
Please see the attachment

Reviewer 2 Report (New Reviewer)
The present manuscript is a very interesting proof of concept for the design of cosmetic formulations. However, some points should be addressed before acceptance:
-Explanation of the values of pH and viscosity as a function of the composition are required. Otherwise, the validity of the work is strongly reduced.
-Abstract should be rewritten in a more concise way
The grammar and spelling is good and only a minor revision is required
Author Response
Reviewer 2: Our group would like to thank the reviewer for his time to reviewer our submitted manuscript and for his fair comments.
The present manuscript is a very interesting proof of concept for the design of cosmetic formulations. However, some points should be addressed before acceptance:
-Explanation of the values of pH and viscosity as a function of the composition are required. Otherwise, the validity of the work is strongly reduced.
Answer: We would like to thank the reviewer for his comment. Two small paragraphs explaining how the composition of the emulsion affects the pH and viscosity have been added in sections 3.1 and 3.2 respectively.
-Abstract should be rewritten in a more concise way
Answer: We would like to thank the reviewer for his comment. The abstract has been revised.
Round 2
Reviewer 2 Report (New Reviewer)
Authors have addressed all the points and now the work is fully publishable
Only minor spelling changes are needed.
This manuscript is a resubmission of an earlier submission. The following is a list of the peer review reports and author responses from that submission.
Round 1
Reviewer 1 Report
I do not see any strong statement to show the novelty of the study. The argument is not new, the work is not original and the methodology as described is not novel. The manuscript is too weak. The novelty and scientific contributions of this manuscript need to be further enhanced and justified. The present form reports a case study. It needs further the scientific in-depth analysis of the results of this paper. An updated and complete literature survey should be conducted. However, the organization of the introduction and the description of the methodology is very poor. In addition, the presentation of the data is incomplete and confusing without any discussion. It is a pain that this novel study cannot be published due to the poor quality of the manuscript. In fact, the works done in this study has the potential to provide very good scientific outputs, however, the organization, representation, and terminological phrasing of the paper are very poor and readers may feel in a mess of work done in the paper. Therefore, As the final review result, the reviewer CAN NOT SUGGEST this paper for publication.
Reviewer 2 Report
This study has a high practical importance, several useful results were gained and showed in the manuscript. However the article in the current for seems to be only a draft version, which is full with errors.
The title is not correct, this study presented not only O/W but also W/O emulsions.
The viscosity and pH stability were measured nicely over storage. The Vit A containing emulsions showed the best sunscreen properties, while Vit C containing emulsions were the best antioxidants.
line 27-29 the third layer of the skin is missing (subcutis or hypodermis)
line 50-51 please check this sentence
line 79 alpha-tocopherol
line 82 stearic acid
line 90 Briefly.....
Table 1 This table seems to be a preliminary draft version, please change it
line 110-111 ??? There are some confusions in this sentence
Stearic acid
Table 2 This is also a draft version, please make the corrections
line 136-138 storage conditions?
line 147-149 storage conditions?
line 179 please be consequent in the use of space 10 min 30 min
No statistical analysis was applied to evaluate the results. Should be performed.
line 192-194 this belongs to the methods section
Figue 1 title Vitamin D???
Figure 6 title???
Figure 7 Title first 8 hours???
line 402 alpha-tocopherol
Discussion and outlook
Cosmetic products are frequently formulated as semisolid formulations. Please add some words about this and about the possible properties of complex formulations (which contain more than one different vitamins). It would worthy to study complex formulations for stability (rheological and pH) and for antioxidant and sunscreen properties.
Reviewer 3 Report
This manuscript the studies about preparation and investigation properties of O/W emulsions containing Vitamins A, C and E. In my opinion, the subject isn't original. Both the composition and the technology of the emulsion aren't original. I do not find the scientific soundness of this study. The use of vitamins A, C, E in cosmetics has been already studied and reported widely in the literature. I don't see anything innovative here. What new knowledge was presented in the article? Regarding the manufacture of emulsions? Or the application of vitamins and the properties they determine? In my opinion, the composition of emulsions is general and the synthetic vitamin substitutes used are too. In addition the manuscript has many shortcomings that made the manuscript unsuitable for publication in present form. The language used is incorrect, it is recommended to read the text by an English-speaking person. The results aren't complete but appropriately described. Units and abbreviations are explicit. The reference materials are well-selected and up to date. The figures not contain all the necessary information, aren’t appropriately captioned but not clear. Discussions and results are based exclusively on the completion of some forms by the subjects, without more evaluation. I would like to point out the following:
1. The subject of the paper is in line with the topic of the journal but is not in line with the content of this article. The title refers to o/w emulsions. After all, also w/o emulsions have been studied. The title only indicates the study of SPF and antioxidant properties at work. Why? After all, rheological tests were also performed.
2. Abstract: line 11: Sentense: In the current work, moisturizing Oil in Water (O/W) and Water in Oil (W/O) emulsions containing Vitamin A, C and E in 0.5, 1 and 2% wt concentrations were prepared. How do you know that these emulsions have a moisturizing effect? Where are the studies confirming this property of these emulsions?
3. Abstract: line 13: . Sentense: The pH and viscosity stability over storage as well as their sunscreen and antioxidant properties of synthesized emulsions were investigated. Synthesized emulsions??? The process of producing an emulsion is not a synthesis. It is a physically process not a chemical reaction.
4. Introduction: line 67. Sentense: Although there are many studies regarding the in vitro release of vitamins from O/W and W/O/W emulsions [22]–[24], to our best knowledge this is the first time vitamins A,C and E have been incorporated and studied in O/W and W/O formulations.
This is not true. Vitamin A in emulsions has been studied. For example:
Moyano, M., & Segall, A. (2011). Vitamin A palmitate and-lipoic acid stability in o/w emulsions for cosmetic application. Journal of Cosmetic Science, 62, 405-415.
Semenzato, A., Bau, A., Dall'Aglio, C., Nicolini, M., Bettero, A., & Calliari, I. (1994). Stability of vitamin A palmitate in cosmetic emulsions: influence of physical parameters. International journal of cosmetic science, 16(4), 139-147.
Carlotti, M. E., Rossatto, V., & Gallarate, M. (2002). Vitamin A and vitamin A palmitate stability over time and under UVA and UVB radiation. International journal of pharmaceutics, 240(1-2), 85-94.
Zasada, M., Budzisz, E., Kolodziejska, J., & Kalinowska‐Lis, U. (2020). An evaluation of the physicochemical parameters and the content of the active ingredients in original formulas containing retinol. Journal of Cosmetic Dermatology, 19(9), 2374-2383.
This is not true. Vitamin E in emulsions has been studied. For example:
Montenegro, L., Rapisarda, L., Ministeri, C., & Puglisi, G. (2015). Effects of lipids and emulsifiers on the physicochemical and sensory properties of cosmetic emulsions containing vitamin E. Cosmetics, 2(1), 35-47.
Wissing, S. A., & Müller, R. H. (2001). A novel sunscreen system based on tocopherol acetate incorporated into solid lipid nanoparticles. International journal of cosmetic science, 23(4), 233-243.
This is not true. Vitamin C in emulsions has been studied. For example:
Gonçalves, G. M. S., & Campos, P. M. (2009). Shelf life and rheology of emulsions containing vitamin C and its derivatives. Revista de Ciências Farmacêuticas Básica e Aplicada, 30(2).
This is not true. Vitamin C in emulsions has been studied. For example:
Akhtar, N., & Yazan, Y. (2008). Formulation and in-vivo evaluation of a cosmetic multiple emulsion containing vitamin C and wheat protein. Pak. J. Pharm. Sci, 21(1), 45-50.
Gonçalves, G. M. S., & Campos, P. M. (2009). Shelf life and rheology of emulsions containing vitamin C and its derivatives. Revista de Ciências Farmacêuticas Básica e Aplicada, 30(2).
Caritá, A. C., de Azevedo, J. R., Buri, M. V., Bolzinger, M. A., Chevalier, Y., Riske, K. A., & Leonardi, G. R. (2021). Stabilization of vitamin C in emulsions of liquid crystalline structures. International Journal of Pharmaceutics, 592, 120092.
5. Materials and Methods . This section should be written so that the experiment can be replicated. This is not the case. The names of the raw materials are missing. Only the chemical names are given. What were the concentrations of raw materials? Some of the names of the compounds used to make the emulsion are incorrectly written. For example: cetostearyl alkohol should be cetearyl alkohol; steatic acid should be stearic acid.
6. Line 87 Fabrication of the O/W and W/O emulsions, better to write: Preparation of the O/W and W/O emulsions.
7. Line 100 - 104. How do we know that W/O and O/W emulsions have been obtained? Did the authors check the type of emulsion using any method? The ratio of aqueous to oil phase by itself does not guarantee a particular type of emulsion.
8. The sections 2.2.1 and 2.2.2 are unnecessary. Everything about the composition of the emulsion is in Table 1.
9. Table 1. The description of the table does not quite relate to its content. It might be better to write: Composition of O/W and w/o emulsions used at work. In addition, the table in its current form is not very clear.
10. Line 147 Who is the viscometer manufacturer? please specify as for ph sensor.
11. Section Result: 3.1 pH stability : line 200: results for emulsions witch vitamin A are missing.
12. Figure 1 caption is incorrect. Did you test for vitamin D? I think not.
13. Where is Figure 2?
14. It is difficult to determine the effect of vitamins on Ph when a pH regulator (citric acid) is used.
15. 3.2 Rheological study. What is the reason for the stability in viscosity for the systems studied? Maybe simply from using Xanthan Gum, which as a polymer is good at stabilising the system? If it were not used, one would see a difference in viscosity for emulsions with different vitamins.
16. Figure 1, 3, 4, 5. In my opinion, points on a graph should not be connected by a line.
17. What the statistical methods were used? Where the error columns in the charts come from?
18. Stability for emulsions should be carried out by load tests, e.g. thermal test, force test (centrifuge test) or on a Zetasizer. Testing pH or viscosity does not guarantee that emulsions are stable.
19. References. item 25: incomplete article data and incorrect formatting. Check the formatting for the full list of references.